# Sensory-Processing Sensitivity and Pathways to Depression and Aggression: The Mediating Role of Trait Emotional Intelligence and Decision-Making Style—A Pilot Study

**DOI:** 10.3390/ijerph182413202

**Published:** 2021-12-15

**Authors:** Nikola Drndarević, Sonja Protić, José M. Mestre

**Affiliations:** 1Institute of Criminological and Sociological Research, 11000 Belgrade, Serbia; sonja.protic@ipu-berlin.de; 2International Psychoanalytic University, 10555 Berlin, Germany; 3University Institute of Social and Sustainable Development (INDESS), University of Cádiz, 11405 Jerez de la Frontera, Spain; 4Department of Psychology, University of Cádiz, 11519 Puerto Real, Spain

**Keywords:** sensory-processing sensitivity, depression, aggression, emotional intelligence, decision-making style, gender differences

## Abstract

While the link between sensory-processing sensitivity (SPS) and internalizing symptoms has been well-established, a link to externalizing problems is still to be explored. This study aimed to further examine the relation between SPS and behavioral problems by testing the potential mediating roles of trait emotional intelligence (TEI) and decision-making styles. Pathway analyses were conducted on data from 268 community sample participants (*M*_age_ = 25.81, *SD* = 2.41, 61.2% females). Results indicated gender differences in the pathway level outcomes of SPS, as well as potential partial mediators in men and women. SPS both directly and via the mediating effects of the well-being factor (TEI) and avoidant decision-making influenced depression, regardless of gender. Direct effects on aggression were, however, obtained only in the male sample. Indirect effects of SPS on aggression were found in spontaneous decision-making for men and in the self-control and sociability factors of TEI for women. Directions for future research were discussed.

## 1. Introduction

Sensory-processing sensitivity (SPS) represents a personality trait characterized by a greater depth of cognitive processing and greater emotional reactivity [1,2]. SPS involves increased biologically-based sensitivity to environmental stimuli [1,2]. Recently, there has been an increased theoretical and practical interest and recognition of the importance of SPS as an essential factor participating in both well-being and difficulties among more sensitive individuals, having implications for health, education, and work [2]. Indeed, SPS could act as an advantage or a disadvantage, depending on whether the environments are supportive or adverse [2,3]. While supportive environments may help more sensitive individuals prosper, adverse environments seem to generate increased negative mental health outcomes, which may be manifested in problem behaviors [3].

Existing research has mainly focused on the relation between SPS and internalizing (INT) problem behavior [4], specifically with depression, anxiety, autism, alexithymia, and somatic problems [5,6,7,8]. The main representative of INT problems, depression [9], has revealed low–moderate positive associations (from 0.22 to 0.35) with more sensitive people [5,6,10]. Neal et al.’s [7] study did not find a correlation between SPS and depression, which may correspond to SPS not being solely a disadvantage.

The established findings of SPS and problem behaviors on the INT spectrum were not surprising, and, indeed, SPS has been characterized by the general avoidance strategy [1,2,3]. However, there are studies showing that greater susceptibility to environmental stimuli may result in both INT and externalizing (EXT) reactions [11]. Should SPS be associated with EXT reactions, this would raise doubt on the general avoidance strategy and pause-to-check behaviors assumed present in those higher in sensitivity [2,3]. Thus, the relationship between SPS and EXT problems remains insufficiently explored. To our knowledge, one study experimentally examined the role of SPS and EXT behaviors in a sample of children, which reported that highly sensitive children showed more EXT behavior problems [11]. No studies have explored yet the association between SPS and EXT problems, and specifically aggression, in older populations.

Notwithstanding the relevance of adverse environments, the mechanisms underlying SPS and psychological distress have been mostly neglected in the literature [2]. More recently, a psychological model of cognitive reactivity has been suggested as a theoretical explanation for depression, anxiety, and somatic problems in more sensitive individuals [12]. Viewed from this model, depression should not be seen as a consequence of stimuli or negative emotions but rather as a secondary phenomenon of cognitive reactivity. In other words, SPS amplifies the negative aspects of situations, stimuli, and cognition increasing the probability of depression symptoms. However, it is not clear how this model would explain the potential relation of SPS to EXT outcomes. Moreover, considering the core facets of SPS (cognitive and emotional processing), it becomes relevant to explore factors belonging to these domains, which may further our understanding of problem behaviors in more sensitive individuals. Thus, it is conceivable that the emotional factor of trait emotional intelligence and cognitive factor of decision-making may function as mediators between SPS and problem behaviors.

### 1.1. Trait Emotional Intelligence as a Mediator between SPS and Problem Behaviors

Emotional intelligence can be conceived and measured differently: as ability (AEI) or as a trait (TEI). AEI refers to accurate emotion perception, emotion generation to assist in thinking, and understanding and reflectively regulating emotions [13]. AEI is assessed using performance tests (e.g., the Mayer-Salovey-Caruso Emotional Intelligence Test, MSCEIT [14]). In contrast, TEI represents a pattern of behavioral dispositions and self-perceptions concerning an individual’s ability to recognize, process, and utilize emotion-laden information [15]. TEI is assessed using self-report scales (e.g., Trait Emotional Intelligence Questionnaire, TEIQue), and consists of four factors [16]: well-being (optimism, happiness, and self-esteem), self-control (impulse control, emotional regulation, and stress management), emotionality (emotional perception and expression and relationship skills), and sociability (social competence, assertiveness, and affecting others’ emotion). The current conceptualization of TEI refers to emotional self-efficacy skills that can be subjected to learning and experience, which can serve as a protective factor against various health and behavioral problems [15]. The choice of TEI instead of AEI was driven by our interest in exploring how individuals perceive their emotional intelligent behaviors rather than emotional abilities themselves. Moreover, there are no satisfactory AEI measures for adolescents [17], and TEI covers social and personal functioning through the subscales [18]. At the global level, the TEIQue-Short Form shows very good precision across most of the latent trait range [18].

Surprisingly, to our knowledge, the TEI construct has not been used in SPS research considering its range of covering one of the core facets of SPS as well as being implicated in various behavioral problems. Indirect research may be used as a guide to such relationship: SPS was related to lower well-being [19]; poor stress management and difficulties in emotion regulation [20]; as well as social and communication deficits [6]. The relationship between emotionality and SPS is ambiguous, considering the mentioned difficulties in emotion regulation, social deficits, and lower self-esteem [3], on the one hand, and higher supposed emotion perception in SPS, on the other [1]. Thus, SPS was expected to affect the TEI factors, especially well-being, self-control, and sociability.

In addition, previous research [21] has indicated a link between TEI and INT (f.i., with depression [22]) and EXT (f.i., aggression [23]) problems. Individual TEI factors were rarely researched, except for one study, which found negative correlations between each TEI factor and depression [22]. Few studies point to mixed results when considering social deviance. Both high and low emotionality, low self-control, and high sociability have been connected with social deviance [24,25]. Nevertheless, the research remains scarce [25], indicating the need for exploration within the mentioned domains.

In sum, the proposed model assumed that SPS directly influences problem behaviors and at least partially through the well-being, self-control, and sociability factors of TEI.

### 1.2. Decision-Making Style as a Mediator between SPS and Problem Behavior

Decision-making styles are conceptualized as learned response patterns of behavior that can be used across contexts and decision situations, and are indicative of the cognitive style of an individual [26]. The authors delineated five decision-making styles: avoidant (avoiding decision-making due to lack of confidence); spontaneous (characterized by a sense of immediacy and impulsiveness); intuitive (iteratively developed reliance on hunches and feelings); rational (systematic and logical approach to situations), and dependent (searching for advice and direction from others).

It has been suggested that SPS might affect decision-making, resulting in enhanced or diminished decision-making [3]. Furthermore, SPS individuals have been characterized as adopting a general avoidance strategy [2,3]. On the other hand, there is a lack of studies concerning decision-making and SPS, whereas, its participation in developing problem behavior or psychopathology is well-documented [27]. Although intuitive decision-making was expected in those higher in SPS [1], it was not clear how this would participate in problem behaviors, and together with rational and dependent decision-making were not the main focus of the study.

Concerning decision-making and problem behaviors, impaired decision-making may function as a factor amplifying the effects toward INT or EXT problems [28]. Previous research has indicated a link between avoidant decision-making and depression [29]. Due to greater sensitivity to risk, the avoidant decision-making style was present in depressive individuals [30]. Conversely, increased risk-taking and impulsive decision-making have been connected with higher levels of aggression [31]. Impulsivity has been connected to decision-making deficits and proneness to aggressive behavior [32], while EXT behaviors were associated with disinhibited decision-making [33].

In sum, the proposed model assumed that SPS directly influences problem behaviors and, at least partially: depression through avoidant decision-making and aggression through spontaneous (impulsive) decision-making.

### 1.3. Role of Gender as a Moderating Factor

Previous research has consistently shown that women score higher on SPS than men [34]. It has been suggested that reported greater sensitivity in women may be due to differences in upbringing and socialization [1]. In comparison to women who report more INT, men report more EXT behavior [35], i.e., greater aggression in men and depression in women was frequently found [9].

Regarding gender and TEI, few studies have examined this relationship with no consensus [21]. A relatively consistent finding was reflected in women scoring higher in emotionality [22] and men scoring higher in self-control and sociability [21]. No difference was registered in the well-being factor [36]. The emotionality factor may lead to a higher prevalence of depressive symptoms (INT) in women [37]. Moreover, EXT outcomes and TEI are moderated with respect to sample, age, and gender. For example, in a sample of male juvenile offenders, lower well-being, self-control, and emotionality was registered with social deviance [24]. In the community sample, both high and low emotionality, low self-control, and high sociability have been connected with social deviance [25]. Additionally, social deviance TEI profiles point to the inconsistency of TEI in the transitional period of adolescence toward adulthood [25], posing further restrictions on prediction.

The literature on decision-making and gender is not in unison. On the one hand, several research studies showed no gender difference in decision-making [26]. On the other hand, women seem to be more averse to risky choices and showed a stronger tendency to adopt an avoidant decision-making style than men [38]. It has been suggested that cultural pressures shape decision-making in men and women, where risky behavior is not favored for women, while the opposite is encouraged for men [39].

### 1.4. Rationale

Previous findings have linked SPS to problem behaviors, predominantly INT behavior [4]. However, to our knowledge, only one study examined environmental sensitivity and EXT behavior [11]. In light of this, we aimed to expand the research of SPS and EXT behavior, specifically aggression. Due to their connections with problem behaviors, TEI and decision-making could act as potential candidates for understanding the relationship between SPS and problem behaviors. Consequently, the present study sought to explore the mediating roles of TEI (emotive factor) and decision-making styles (cognitive factor), in the relationship of SPS and depression and aggression.

Based on previous research, we constructed a model (Figure 1) where: (1) SPS is associated with and is directly influencing problem behaviors, both depression [5,6,7] and aggression [11], (2) SPS is indirectly influencing problem behaviors, reflected in the partial mediating roles of TEI [16] and decision-making styles [3]. More specifically, we hypothesized that the well-being factor of TEI partially mediates SPS and depression [3,4,20], while self-control and sociability partially mediate SPS and aggression [24]. As previously argued, the findings in the literature concerning emotionality and well-being in relation to aggression are ambiguous, warranting further exploration. Moreover, regarding decision-making styles, a factor differentiating a pathway toward depression is reflected in avoidant and a pathway toward aggression in spontaneous decision-making. Finally, we were interested in exploring the potential moderating influences of gender for the proposed pathways.

## 2. Materials and Methods

### 2.1. Participants

The study included a total of 268 participants that fully completed the questionnaire packet (61.2% (n = 164) were women). The respondents age varied from 17 to 30 years with a mean age of 25.81 years (Median = 26, Mode = 26, *SD* = 2.41). Education was represented by three levels: 1. Middle or high school (18.3%); 2. Bachelor studies (44%); 3. Masters studies (37.7%).

### 2.2. Procedure

The study was approved by the Ethics Commission of the Faculty of Philosophy, University of Belgrade, and was conducted with the help of an online platform, using convenience and snowball sampling techniques. The time estimated for questionnaire completion was between 15 and 25 min. In order to ensure more significant variability in scores, respondents from social networks and forums for people with higher sensitivity were also recruited. Respondents participated voluntarily, and the standards of informing, confidentiality, anonymity, and data retention were met.

### 2.3. Instruments

The Highly Sensitive Person Scale (HSPS [1]). The HSPS consisted of a 27-item scale in which participants were asked to rate their agreement with various statements on a Likert Scale ranging from 1 (strongly disagree) to 7 (strongly agree). Items reflected sensitivity to both internal and external stimuli (e.g., “Are you made uncomfortable by loud noises?”). Recent research suggests SPS being a continuous trait [2]. Higher scores indicated greater sensory-processing sensitivity. The test items were adapted from English to Serbian using the double-translation procedure.

General Decision-Making Style (GDMS [26]). The GDMS consisted of 25 items in which participants were asked to rate the items on a five-point Likert-type scale (e.g., “I make quick decisions.”). The items measured five decision-making styles (rational, dependent, intuitive, avoidant, and spontaneous), each consisting of five items. The test items were adapted from English to Serbian using the double-translation procedure.

Trait Emotional Intelligence Questionnaire-Short Form (TEIQue-SF [16]; the Serbian version of the Trait Emotional Intelligence Questionnaire [40]). The TEI was assessed via the short form of TEIQue, consisting of 30 statements responded to on a seven-point Likert scale (e.g., ‘I often find it hard to understand other people’). Higher scores indicated a higher level of TEI. The test measured four factors (well-being, self-control, emotionality, and sociability).

Beck Depression Inventory-II (BDI-II [41]; the Serbian version of the BDI II [42]. The BDI-II is a 21-item self-report instrument reflecting cognitive, affective as well as somatic aspects of depression. Higher scores suggested symptoms of depression.

Reactive–Proactive Aggression Questionnaire (RPAQ [43]). The RPAQ is a 23-item measure, assessing aggression on a three-point Likert scale ranging from 0 (never) to 2 (always) (e.g., “How often have you used physical force to get others to do what you want”). In this study, we focused on the total aggression score. Higher scores indicated higher scores of aggression. The test items were adapted from English to Serbian using the double-translation procedure.

### 2.4. Data Analysis

Primary analyses were performed using SPSS for Windows (version 26.0, SPSS Inc., Chicago, IL, USA). Descriptive statistics illustrated means and standard deviation scores on measured variables in the total sample, while correlations were tested using Pearson’s correlation coefficient. The independent sample t-test was used to test group differences. Path Analysis (in AMOS, version 24) was conducted to test the theoretical model that predicts depressive or aggressive behavior from SPS with the emotional intelligence and decision-making styles as mediators. Specific indirect effects were estimated using the AMOS plugin [44]. Estimates of indirect effects were made using a bias-corrected bootstrap technique with 2000 samples and 95% confidence intervals.

## 3. Results

Means, standard deviations, and score ranges, as well as internal consistencies for all examined variables, are shown in Table 1. Cronbach’s alphas were mostly adequate and comparable to those obtained in previous research.

Statistically significant correlations were in expected directions: SPS was positively correlated with depression (r = 0.44, *p* < 0.01), aggression (r = 0.21, *p* < 0.01), avoidant (r = 0.26, *p* < 0.01) and intuitive (r = 0.21, *p* < 0.01) decision-making, while negatively with three out of four TEI factors: well-being (r = −0.26, *p* < 0.01), self-control (r = −0.37, *p* < 0.01), and sociability (r = −0.31, *p* < 0.01). No correlation was registered between SPS and the emotionality factor as well as between SPS and spontaneous decision-making.

Path analysis was utilized to test the theoretically assumed model, which was corrected following obtained correlations. The path from SPS to depressive and aggressive types of reacting was outlined. Among the factors constituting TEI, as mediators were chosen: well-being, self-control, and sociability since they showed correlations with both SPS on the one hand and either depression or aggression on the other hand. Emotionality was not included as a mediator due to the lack of significant correlation. The avoidant decision-making style was added as a mediator for SPS and depression and spontaneous decision-making as a mediator for SPS and aggression. Although, intuitive decision-making showed correlations to SPS and aggression, and spontaneous decision-making did not correlate with SPS, the final model showed a better fit with the exclusion of intuitive and inclusion of the spontaneous decision-making style. Finally, the modified model (Figure 2) resulted in an acceptable fit: χ2 (5) = 9.253, ns; TLI = 0.953; CFI = 0.992; RMSEA = 0.056.

SPS showed the following patterns of influence (Table 2): negative effects on well-being, self-control, and sociability; positive effects on avoidant and spontaneous decision-making; and positive effects on depression and aggression. Among the TEI factors: well-being negatively influenced depression but not aggression; self-control exerted a negative influence on aggression but not on depression; sociability positively influenced aggression, but no effect was registered for depression. Out of the decision-making styles, spontaneous decision-making positively affected aggression; while avoidant decision-making positively affected depression. Finally, partial mediators were revealed for depression: well-being (β = 0.102, *p* < 0.001) and avoidant decision-making (β = 0.043, *p* < 0.01). For aggression: self-control (β = 0.084, *p* < 0.01).

Regarding the gender differences, men registered higher scores in aggression (M_men_ = 11.81 and M_women_ = 9.87; t (266) = 2.54, *p* < 0.05) and self-control (M_men_ = 4.93 and M_women_ = 4.42; t (255) = 4.00, *p* < 0.001), while women had higher scores in SPS (M_women_ = 4.73 and M_men_ = 4.26; t (266) = −4.84, *p* < 0.001), depression (M_women_ = 10.97 and M_men_ = 7.48; t (266) = −3.52 *p* < 0.001), emotionality (M_women_ = 5.26 and M_men_ = 5.01; t (266) = −1.98, *p* < 0.05), and intuitive decision-making (M_women_ = 3.81 and M_men_ = 3.02; t (266) = −3.13, *p* < 0.01).

Compared to the total sample, inspection of Pearson’s correlations for men and women revealed several differences (Table 3). In the sample of men, SPS positively correlated with aggression (moderate) and spontaneous decision-making (small). On the other hand, in the sample of women SPS negatively correlated with well-being (moderate) and positively with intuitive decision-making (small).

Testing for gender invariance showed the model as gender invariant (χ^2^ (13) = 20.066, *p* > 0.05), meaning that the differences between men and women were not registered on the modal level. However, differences were registered on the path level (Figure 3). The positive relationship between SPS and spontaneous decision-making (z = −1.23, *p* > 0.05) and SPS and aggression (z = −2.91, *p* < 0.001) were only significant for men. Furthermore, the positive relationship between sociability and aggression was only significant for women (z = 0.9, *p* > 0.05). Finally, the negative relationship between well-being and depression was stronger for women (z = −2.12, *p* < 0.01).

The estimates for both models are shown in Table 4. SPS was associated with and directly influenced problem behaviors, both depression (regardless of gender) and aggression (only for men). SPS indirectly influenced problem behaviors through mediating roles of TEI and decision-making styles. Specifically, partial mediation was registered for wellbeing and depression (regardless of gender), avoidant decision-making and depression (regardless of gender), spontaneous decision-making and aggression (only for men), and self-control, sociability, and aggression (only for women).

## 4. Discussion

The present study aimed to explore the pathway model from sensory-processing sensitivity toward depression and aggression as the INT and EXT manifestations of distress. The mediating roles of TEI and general decision-making style, in addition to gender differences, were tested in different pathway models. The construed model excluded the emotionality factor of TEI due to its lack of correlation with SPS. It is possible that emotionality includes some aspects at which high SPS individuals are assumed good at (e.g., empathy) and other aspects in which they may be subpar (e.g., relationship skills), thereby conceivably resulting in no correlation. However, for a clearer picture it would be necessary to repeat the analyses on a larger sample.

The suggested theoretical explanation for the underlying mechanism behind SPS and depression was a secondary cognitive reactivity [2,12]. The authors argued that SPS was a sole factor responsible for amplification of negative sensory stimuli, resulting in secondary phenomenon of cognitive reactivity (maladaptive content and processes), which in turn leads to depression, anxiety, and psychosomatic problems [12]. However, the provided model overvalues cognitive at the expense of emotive factors, and does not offer an explanation for the other type of problem behaviors—EXT behavior. Building on this model, we proposed that the negative stimuli amplification in both SPS core facets (cognitive and emotive processing), results in psychological distress, which may consequently be manifested in INT (depressive) or EXT (aggressive) behavior. In the following paragraphs, we discuss the potential factors explored in this study from both cognitive (decision-making styles) and emotive (TEI) domains that may participate in psychological distress and problem behaviors.

Before moving on to the questions regarding the model, brief comments are provided for research variables and their correlations. The results from mean score differences and correlations mostly corresponded to previous research.

Firstly, in line with previous studies and as expected, men scored higher in aggression [9] and self-control [21], and women scored higher in SPS [34], depression [9], and emotionality [22], while no difference was found in well-being [36]. Sociability showed no difference, which differed from previous studies [21]. No differences in avoidant or spontaneous decision-making were registered, which supported the studies that found no gender differences in decision-making styles [26], with the exception of intuitive decision making, which was found to be greater in women in this study.

Secondly, the expected pattern of associations was found in: SPS with depression [5]; and aggression [11]; well-being [22], self-control [22], and sociability [22], but not in emotionality; avoidant and intuitive decision-making. Inspection of associations in the sample of men and women revealed important differences. SPS correlated with aggression and spontaneous decision-making only in men, while SPS showed association with well-being only in women. In light of the found gender differences, the interpretations were focused at the path level analysis.

### 4.1. Sensitivity and Depression: Roles of TEI and Decision-Making Style

Sensitive individuals were more susceptible to depression, which is in line with previous studies [5,6,10]. The direct path from SPS toward depression still leaves the possibility of the inbuilt qualities of sensitive individuals to process stimuli more deeply. This result would support recent theories of depression as a secondary phenomenon of cognitive reactivity (i.e., maladaptive thought content and processes) to sensory information and related negative emotions [12]. However, the indirect path through well-being acted as an added factor, which was especially prominent in influencing depression in women, pointing to the relevance of the emotional factor. It could be that while negative emotions possess a greater significance, in highly sensitive people such emotions are greatly amplified and may trigger psychological distress. This increased emotional reactivity could be then reflected in a less positive/optimistic representation of themselves, their past, present, and future as well as greater unhappiness and lower self-esteem (represented in lower well-being). These results conform to reports on depression [22], supporting Beck’s notion that low self-regard is the core factor in depression [41]. Such excess in negative emotions may engage greater focus on negative emotions leading to ruminative strategies and emotional decontrol favoring the internalizing behavior [9]. Despite SPS being associated in this and previous studies with poor stress management and difficulties in emotion regulation [20], as well as with social and communication deficits [6], self-control and sociability were not found to be mediators for depression in this sample.

Expectedly, avoidant decision-making acted as a partial mediator for SPS and depression, pointing to the possible differentiating effect on depression in contrast to an aggressive type of behavior. Being more sensitive seemed to have effects on the cognitive style of avoidant decision-making, which may have led to the mentioned problem behaviors, pointing to possible directionality of decision-making on depression, rather than the reverse [27]. This may reflect a greater sensitivity to risk present in depressed individuals [29,30], which may have been emphasized in more sensitive individuals. It was also in line with the general avoidance strategy assumed to be present in more sensitive individuals [1]. These findings lend tentative support to our proposition of both emotional and cognitive factors participating in problem behaviors. Being more sensitive may amplify negative stimuli and overload emotive and cognitive capacities, slowing or inhibiting decision-making.

### 4.2. Sensitivity and Aggression: Roles of TEI and Decision-Making

A novel finding was reflected in SPS being associated with and influencing aggression, limited to men. This pathway raises questions about pause-to-check behaviors and the general avoidance strategy assumed to be present in more sensitive individuals [1]. In contrast to women, SPS presented as a risk factor for aggression in the direct path in men. This result points to different strategies in dealing with psychological distress, with the tendency of men to choose EXT and the tendency of women to choose INT strategies [9,35].

Further differences in aggression in men and women were found when considering the indirect effects. A sense of immediacy and impulsiveness (reflected in spontaneous decision-making style) acted as a partial mediator in men but not in women. These results were in line with the research supporting using different coping strategies when in distress [9,45]. In comparison to men, women were found to be more focused on mental states, both their own and others’, and therefore engaged in aggression for reasons different than men [46]. Women seem to have a greater tendency to choose emotion-focusing while men seem to have a tendency to suppress or avoid dealing with emotions [9], which corresponds to greater INT problems found in women and greater EXT problems found in men [35]. Greater emotion-focusing in women also corresponded to significantly greater effect of the well-being factor in the depression pathway in sensitive women found in this study. Conversely, men seem to engage in aggression impulsively and without considering mental states [46], which corresponded to EXT problems of suppressing and avoidance of emotion previously mentioned [9]. Avoidant and spontaneous decision-making styles, supported the idea of impaired decision-making amplifying the effects toward INT or EXT problems [28].

In partial accordance to previous studies [24], lower emotion regulation and impulse control (reflected in lower self-control) and greater social competence and affecting the emotions of others (reflected in higher sociability) was predictive of aggression, although only in women. Lower self-control has resemblance to impulsivity, which has been previously connected to aggression [32]. Greater sociability may be suggested to have been utilized for attaining and maintaining social status [45]. An interesting relationship emerged with SPS negatively affecting self-control and sociability [20], and while self-control negatively affected aggression, sociability positively affected aggression in women. Being a more sensitive woman seems to lower sociability thereby possibly acting as a suppressive factor for aggression in women. These results seem to point to different TEI profiles with respect to gender [25]. However, the interpretation of this finding is restricted by both the lack of research in this area [25] and the sample size of the current study. A larger sample may point to the importance of such factors in men as well.

### 4.3. Practical Implications

Considering the pilot nature of the study, the theoretical and practical implications should be offered cautiously. Should these results be replicated, it would contribute to the theory on SPS participating in problem behaviors to include the potential of externalizing reacting, as a way to dispense with negative emotions. Rather than opposing the original theory, we propose it caters to the complexity of the phenomena. More specifically, although the study results support the main idea of highly sensitives adopting the avoidant strategy [1,3], it should account for the possibility of adopting the impulsive strategy as well. Moreover, externalizing reacting seems to be influenced by gender, further suggesting the need for both theoretical and practical considerations of gender differences.

Drawing from a proposed model of SPS and psychological distress offered in this study, we suggest two points that can be used to base practical interventions. The first point refers to SPS having a negative effect on well-being, self-control, and sociability as well as on increasing the potential for avoidant and spontaneous decision-making. Considering the increased benefits highly sensitive persons receive from positive interventions [2], one relevant way may be through a supportive person with average sensitivity (acting as external regulating agent) to regulate distress and reduce the overwhelming stimuli, thereby aiding the highly sensitive person to cope with disconcerting stimuli [47]. The second point refers to the importance of emotional reactivity and specifically negatively impacted well-being in highly sensitive persons. A potential suggestion for diminishing emotional reactivity would encompass improvement of self-efficacy skills in regulating emotions [20], as well as Mindfulness-Based Cognitive Therapy [2,12]. Highly sensitive persons may also benefit from physical exercises, which seem to moderate the association with depression [10].

### 4.4. Limitations and Future Directions

Firstly, this study was conducted on a community sample, limiting generalizability and suggesting the need for a clinical sample with respect to psychological distress and problem behaviors. Further restrictions were posed by a relatively small sample of men. Moreover, significant effects with specified directions obtained using path analysis for testing do not exclude the possibility of other good-fitting alternative models with differently proposed links and directions and thus imposing limits to causal interpretations. Longitudinal research would be adequate for this aim. Construal of the model in this study was guided by the suggested theoretical model of SPS and psychological distress [12] with the choice of cognitive and emotive variables as core facets of SPS, which were pertinent to problem behaviors.

An impetus for further research avenues may thus be twofold. Considering the supportive evidence gained from this pilot study, further cognitive and emotive factors warrant exploration in the relation of SPS and problem behaviors. Moreover, although we were particularly interested in traits, future research could extend our knowledge by including (instead) the ability measures of emotional and cognitive functioning. Although the application of self-report measures faces well-known biases and limitations, TEIQue covers representative aspects of socioemotional functioning, and even the short form showed good psychometric characteristics [18]. Replication of these findings on a broader sample, including other age groups (such as children or older adults), is also advised. Additionally, such models are adequate at explaining the bottom-up processing (from the environmental stimuli to cognitive and emotional factors), but may be limited at explaining the top-down processing (from cognitive and emotional factors toward stimuli) [12]. Not all sensitive individuals end up in a disadvantageous mental state [1,2,3]. The question of supportive and adverse environments should be reintroduced and expanded beyond early parental [48], and toward cultural contexts, with the inclusion of gender differences. A potentially relevant variable may also be found in a personality structure of the individual.

## 5. Conclusions

Despite limitations, this study is the first to extend SPS research to the externalizing type of reacting through aggression. Moreover, representing the core facets of SPS, the study tested both emotional (TEI) and cognitive (decision-making styles) mediators considered to be relevant and participating in problem behavior. Gender differences are also highlighted. In essence, this study suggests that higher sensitivity coupled with distress may manifest in both depression (INT) and aggression (EXT). With respect to depression, both the emotive factor of well-being and a cognitive factor of avoidant decision-making participated as partial mediators for depression, regardless of gender. Self-control and sociability did not participate as mediators for depression. Concerning aggression, only self-control participated as a partial mediator. Additionally, gender differences shed more light in the pathways of aggression. SPS was directly associated with aggression only in men. Indirect paths also differed, with spontaneous decision-making (characterized by impulsiveness and urgency) showing partial mediation only for men, while self-control and sociability showed partial mediation only for women. This finding corroborates different reasons behind aggression in men and women. These results are in line with the theoretical proposition that SPS acts as an amplifier of negative stimuli, resulting in diminishing of the both emotional and cognitive capacities of the individual, possibly responsible for problem behaviors, both depression and aggression.

## Figures and Tables

**Figure 1 ijerph-18-13202-f001:**
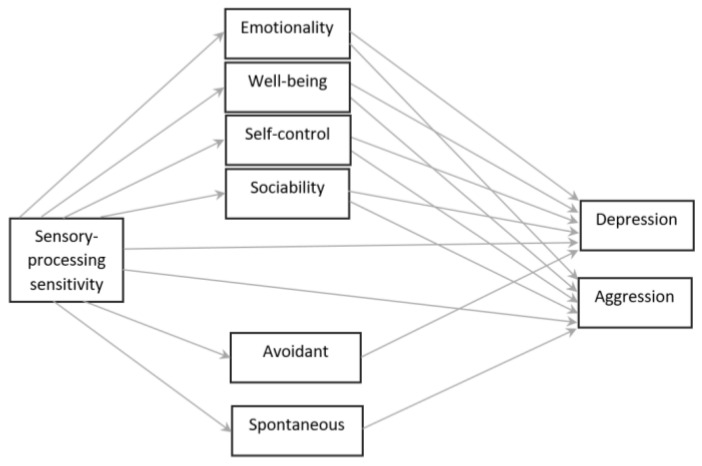
Theoretical model of sensory-processing sensitivity and pathways to depression and aggression: mediating roles of trait emotional intelligence and decision-making styles.

**Figure 2 ijerph-18-13202-f002:**
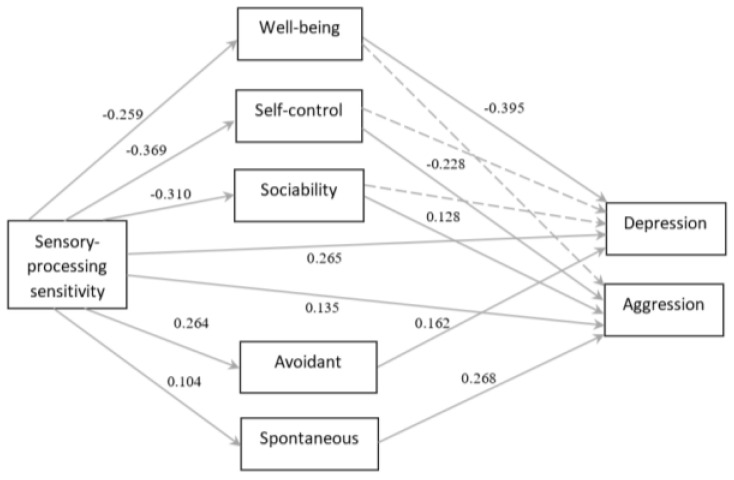
Results for the path analysis on the total sample. Entries are standardized regression weights. Discontinuous lines represent nonsignificant paths. Non-Normed Fit Index TLI = 0.953; Comparative Fit Index CFI = 0.992; root mean square error of approximation RMSEA = 0.056; chi-square χ2 (5) = 9.253, *p* > 0.05.

**Figure 3 ijerph-18-13202-f003:**
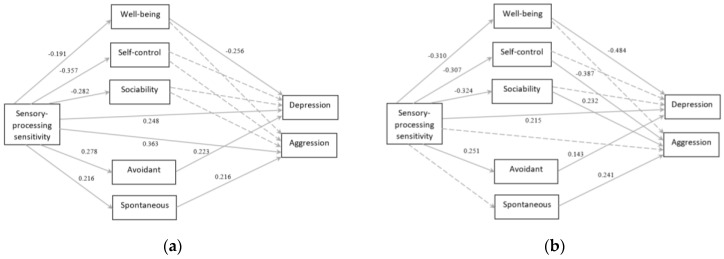
Results for the path analysis for: (**a**) men; (**b**) women. Entries are standardized regression weights. Discontinuous lines represent non-significant paths.

**Table 1 ijerph-18-13202-t001:** Descriptive statistics and internal consistencies among variables.

Variables	α	*M* (*SD*)	Min.–Max
SPS	0.88	4.55 (0.81)	2.11–6.44
Depression	0.90	9.62 (8.08)	0.00–52.00
Aggression	0.86	10.62(6.15)	0.00–46.00
Well-being	0.81	5.41 (1.01)	2.50–7.00
Self-control	0.73	4.62 (1.06)	1.33–7.00
Emotionality	0.70	5.16 (0.98)	2.13–7.00
Sociability	0.69	4.72 (0.95)	2.00–7.00
Spontaneous	0.83	2.47 (0.91)	1.00–5.00
Rational	0.79	4.08 (0.66)	1.80–5.00
Avoidant	0.87	2.79 (1.05)	1.00–5.00
Intuitive	0.86	3.69 (0.82)	1.17–5.00
Dependent	0.81	3.61 (0.80)	1.40–5.00

**Table 2 ijerph-18-13202-t002:** Paths, unstandardized regression weights, and standardized errors in total sample.

**Direct Effects**			
**Paths**	**B**	**S.E.**	**95% CI**
SPS → Well-being	−0.323 ***	0.07	−0.472, −0.190
SPS → Self-control	−0.482 ***	0.07	−.625, −0.337
SPS → Sociability	−0.362 ***	0.07	−0.503, −0.233
SPS → Avoidant	0.344 ***	0.08	0.192, 0.495
SPS → Depression	2.645 ***	0.50	1.732, 3.757
SPS → Aggression	1.026 *	0.46	−0045, 2.183
Well-being → Depression	−3.159 **	0.42	−4.142, −2.251
Self-control → Depression	−0.672	0.43	−1.643, 0.211
Self-control → Aggression	−1.326 ***	0.40	−2.119, −0.403
**Indirect Effects**			
**Paths**		**B**	**95% CI**
SPS → Well-being → Depression		1.022 ***	0.615, 1.606
SPS → Avoidant → Depression		0.427 **	0.185, 0.765
SPS → Self-control → Aggression		0.639 **	0.279, 1.057

Note. * *p* < 0.05; ** *p* < 0.01; *** *p* < 0.001.

**Table 3 ijerph-18-13202-t003:** Correlations among study variables.

Variables	1	2	3	4	5	6	7	8	9	10	11	12
1. SPS	---	0.42 **	0.12	−0.31 **	−0.31 **	0.04	−0.32 **	0.05	0.01	0.25 **	0.16 *	0.13
2.Depression	0.40 **	---	0.22 **	−0.61 **	−0.39 **	−0.22 **	−0.43 **	0.13	−0.05	0.36 **	0.02	0.14
3.Aggression	0.45 **	0.41 **	---	−0.11	−0.36 **	−0.15 **	−0.01	0.35 **	−0.11	0.28 **	0.29 **	0.11
4.Well-being	−0.19	−0.40 **	−0.20 *	---	0.44 **	0.26 **	0.51 **	0.01	0.15	−0.29 **	0.16 *	−0.10
5.Self-control	−0.36 **	−0.43 **	−0.38 **	0.50 **	---	0.20 *	0.52 **	−0.31 **	0.40 **	−0.43 **	0.01	−0.27 **
6.Emotionality	−0.05	−0.14	−0.29 **	0.18	0.29 **	---	0.36 **	−0.15	0.13	−0.36 **	0.18 *	0.11
7.Sociability	−0.28 **	−0.34 **	−0.10	0.36 **	0.34 **	0.45 **	---	−0.08	0.30 **	−0.50 **	0.10	−0.14
8.Spontaneous	0.22 *	0.26 **	0.37 **	−0.10	−0.42 **	−0.21*	−0.06	---	−0.49 **	0.22 **	0.45 **	−0.12
9.Rational	−0.04	−0.18	−0.21 *	0.33 **	0.42 **	0.29 **	0.11	−0.53 **	---	−0.17*	−0.15	0.13
10.Avoidant	0.28 **	0.38 **	0.18	−0.15	−0.38 **	−0.13	−0.26 **	0.26 **	−0.26 **	---	0.02	0.28 **
11.Intuitive	0.18	0.16	0.070	0.28 **	−0.15	−0.02	0.05	0.58 **	−0.06	0.10	---	0.00
12.Dependent	0.06	0.02	−0.10	0.21*	−0.17	−0.02	−0.12	0.09	0.04	0.30 **	0.17	---

Note. Men below diagonal and women above. * *p* < 0.05; ** *p* < 0.01.

**Table 4 ijerph-18-13202-t004:** Paths, unstandardized regression weights, and standardized errors for men and women.

Men				Women			
**Direct Effects**							
**Paths**	**B**	**S.E.**	**95% CI**	**Paths**	**B**	**S.E.**	**95% CI**
SPS → Well-being	−0.211 *	0.11	−0.358, −0.030	SPS → Well-being	−0.438 ***	0.11	−0.607, −0.270
SPS → Self-control	−0.422 ***	0.11	−0.643, −0.218	SPS → Self-control	−0.432 ***	0.11	−0.613, −0.261
SPS → Sociability	−0.334 **	0.11	−0.502, −0.156	SPS → Sociability	−0.399 ***	0.09	−0.562, −0.246
SPS → Avoidant	0.354 **	0.12	0.151, 0.555	SPS → Avoidant	0.353 ***	0.11	0.162, 0.527
SPS → Spontaneous	0.239 **	0.11	0.026, 0.452	SPS → Spontaneous	0.064	0.10	−0.090, 0.213
SPS → Depression	2.095 **	0.79	0.984, 3.301	SPS → Depression	2.401 ***	0.70	1.304, 3.574
SPS → Aggression	3.208 ***	0.79	1.420, 5.225	SPS → Aggression	0.436	0.53	−0.389, 1.245
Well-being → Depression	−1.951 **	0.71	−3.282, −0.597	Well-being → Depression	−3.813 ***	0.52	−4.800, −2.848
Self-control → Depression	−0.884	0.73	−2.025, 0.103	Self-control → Depression	−0.390	0.55	−1.515, 0.609
Self-control → Aggression	−1.750 **	0.59	−2.960, −0.376	Self-control → Aggression	−1.955 ***	0.44	−2.789, −1.212
Sociability → Aggression	1.048 *	0.48	0.111, 1.981	Sociability → Aggression	1.337 **	0.48	0.529, 2.111
Spontaneous → Aggression	1.534 **	0.55	0.464, 2.690	Spontaneous → Aggression	1.412 ***	0.43	0.610, 2.230
**Indirect Effects**							
**Paths**		**B**	**95% CI**	**Paths**		**B**	**95% CI**
SPS → Well-being → Depression		0.412 *	0.071, 1.008	SPS → Well-being → Depression		1.672 ***	0.956, 2.620
SPS → Avoidant → Depression		0.524 **	0.158, 1.225	SPS → Avoidant → Depression		0.399 *	0.092, 0.976
SPS → Spontaneous → Aggression		0.413 *	0.089, 1.030	SPS → Self-comtrol → Aggression		0.844 ***	0.455, 1.441
				SPS → Sociability → Aggression		−0.534 **	−0.996, −0.230

Note. * *p* < 0.05; ** *p* < 0.01; *** *p* < 0.001.

## Data Availability

The database is available on request by the first author.

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
