# Peer review of "Sensory-Processing Sensitivity and Pathways to Depression and Aggression: The Mediating Role of Trait Emotional Intelligence and Decision-Making Style—A Pilot Study"

_ijerph, 2021, doi:10.3390/ijerph182413202_

Round 1

Reviewer 1 Report

The manuscript is clear in both its aims and applied methodology. It offers  relevant findings for both future research and clinical work.

I do not have particular comments to offer as I agree with the discussion  contents.

What is the main question addressed by the research?

The author examine the relation between the sensory-processing sensitivity (SPS) and behavioural problems by testing the potential mediating roles of trait emotional intelligence (TEI) and decision-making styles.

More precisely, they answer to the following questions:

- Which traits of TEI and decision-making style reveal connections with both sensory-processingsensitivity (SPS) and and behavioural problems? 

-  Which traits of TEI among well-being, self-control and sociability can mediate the impact of SPS on depression and aggression, as behavioural problems?

- How spontaneous decision-making style can modulate the effect of SPS on depression?

- How avoidant decision-making can affect SPS and therefore modulate the aggression behavioural tendency? 

- Do trait emotional intelligence (TEI) and decision-making styles show any differences between males and females? In other words, can gender influence depressive and aggressive problems?

Is it relevant and interesting? 

Yes, in my modest opinion, this paper is interesting and relevant because the data analysis shows which traits of TEI and styles of decision-making correlate with both SPS and either depression or aggression and each not (like emotionality and intuitive decision-making).

The study also reports the positive and negative effects of the mediators in mitigate or exacerbate the effect of sensitivity on depressive and aggressive problems. These findings are relevant in therapy. From a research point of view, the study examines the relationship between SPS and external problems like aggression, while the majority of studies evaluated only internal problems (like depression).

How original is the topic?

The paper is original because it considers the psychological model of cognitive reactivity to explain the potential relation of SPS to both internal and external problem behaviours. The model has already suggested how depression, in more sensitive individuals, should be seen as a secondary phenomenon of cognitive reactivity. The authors expand the investigation by considering the SPS, which amplifies the negative aspects of situations, stimuli, and cognition, not only in relation of depression symptoms but also in potential relation of SPS to external outcomes (aggression).

What does it add to the subject area compared with other published material? 

I appreciate the statistical method (path analysis in AMOS) to test the theoretical model that predicts depressive or aggressive behaviour from SPS with the emotional intelligence and decision-making styles as mediators.

The authors offer a clear lecture of the results.

Is the paper well written?

I am not a native English speaker but I read easily the paper.

Is the text clear and easy to read? 

Yes it is.

Are the conclusions consistent with the evidence and arguments presented? 

Yes.

Do they address the main question posed?

Yes, they did.

Author Response

We thank Reviewer 1 for encouraging feedback.

Reviewer 2 Report

Dear colleagues, I hope this message find you well.

Thank you for giving me the opportunity of reading the work “Sensory-processing sensitivity and pathways to depression and aggression: the mediating role of trait emotional intelligence and decision-making style. A pilot study., it has been a very big pleasure to collaborate reviewing this manuscript. The topic of this paper is very interesting and it seems necessary to delve it. However, there are some minor issues to improve before to publish it: 

Introduction

  • The structure of the introduction is very clear, congratulations.
  • Considering the specificities of the journal - which has a multidisciplinary readership - I strongly recommend the authors to introduce why EI has been following the trait approach.

Method

  • It is ok. However, in my opinion, you can add a table describing more in detail the sample characteristics.

Results

  • Ok

Discussion

  • I agree with the limitations detailed, but I’m sure that you can add more specific issues to address in future research. For instance, the several bias and limitations of this kind of instruments. Why did you choose the TEIQue?
  • In my humble opinion, it could be useful to describe in more detail the practical and theoretical implications of this research. It would be useful they contextualize better the contribution within the framework of the issue explaining why the contribution is useful and enrich the impact.

Conclusions

  • Ok

Author Response

Thank you for both the positive feedback and for sharing your suggestions.

Considering the specificities of the journal - which has a multidisciplinary readership - I strongly recommend the authors to introduce why EI has been following the trait approach. 

ANSWER: We appreciate this reflection. We introduced both the specifics and differences between the ability and trait EI as well as argumentation for choosing the trait EI approach (page 2, Introduction part)

It is ok. However, in my opinion, you can add a table describing more in detail the sample characteristics.

ANSWER: We do agree that tables are often a better option for presenting the sample characteristics. However, we do not have data beyond age, education and gender, that we already introduced in the text. If you prefer this data in the table, we could make it.

I agree with the limitations detailed, but I’m sure that you can add more specific issues to address in future research. For instance, the several bias and limitations of this kind of instruments. Why did you choose the TEIQue?

ANSWER:  We added the argumentation for choosing TEIQues and discuss further the limitations on pages 12 and 13.

In my humble opinion, it could be useful to describe in more detail the practical and theoretical implications of this research. It would be useful they contextualize better the contribution within the framework of the issue explaining why the contribution is useful and enrich the impact.

ANSWER: Thank you for pointing this out. We added a whole new section on practical implications (page 12).

Reviewer 3 Report

Congratulations to the authors for their approach to research and the opportunity to address this theme of study, of great investigative relevance to clarify and explain the mechanisms through which the relationship between SPS and psychological adjustment operates (in a similar way, symptoms are addressed internalizing and externalizing) for a show for teenagers, young people and adults. As suggestions for future studies on the same theme, the sample should be expanded to prove that these first results are maintained in the same sense, so it is suggested to center the sample on a similar evolutionary period or to control the age in statistical analyses. I would like to know if you have controlled this variable in this study or if you have bold differences. As for improvement suggestions, the following are proposed.

Introduction:
- Include a figure with the hypothesized model.
Results:
- Line 170: Replace the point and eat by point and end the sentence in the paragraph.
- Line 185: Include after the point in the citation and review these formal questions in the entire text.
- Revise table 1 in terms of format, superimpose the letters in the first column in the revised version.
Discussion:
- Line 308: Indicate that it would be necessary to continue investigating this result with muetras more magnified. Highlight the result referring to the health dimension of the TEI, which plays a particularly relevant role in depression. Interpret this data.
- Highlight the implications for professional intervention.

Author Response

We thank Reviewer 3 for this positive feedback.

As suggestions for future studies on the same theme, the sample should be expanded to prove that these first results are maintained in the same sense, so it is suggested to center the sample on a similar evolutionary period or to control the age in statistical analyses. I would like to know if you have controlled this variable in this study or if you have bold differences.

ANSWER: We did not control for age since we used a relatively homogeneous group of people recently recognised as the transitional age youth (TAY) (Martel & Fuchs, 2017; Vogel & Rosner, 2020). This age range seemed to be too short to discuss the age influence on emotional and cognitive traits. However, we do find this issue relevant and we emphasized the importance of testing our model in the greater sample and other age groups, such as children and adults (page 12, Limitations and future direction).

Include a figure with the hypothesized model.

Done.

Line 170: Replace the point and eat by point and end the sentence in the paragraph.
Line 185: Include after the point in the citation and review these formal questions in the entire text.

Corrected.

Revise table 1 in terms of format, superimpose the letters in the first column in the revised version

We are not sure we understood the suggestion, however, we did correct the table format by adding bold font for all table headings and the first column. If there is anything else, we are open to further suggestions.

Line 308: Indicate that it would be necessary to continue investigating this result with muetras more magnified.

True. Done.

Highlight the result referring to the health dimension of the TEI, which plays a particularly relevant role in depression. Interpret this data.

We are now offering an interpretation on pages 10 and 11, in the Discussion part.

Highlight the implications for professional intervention.

Thank you for this comment - the same was suggested by Reviewer 2 and we agree that practical implication would strengthen our manuscript. Thus, we added a whole new section on practical implications (page 12).
